# RoboView-Bias: Benchmarking Visual Bias in Embodied Agents for Robotic Manipulation

## Abstract

The safety and reliability of embodied agents rely on accurate and unbiased visual perception. However, existing benchmarks mainly emphasize generalization and robustness under perturbations, while systematic quantification of visual bias remains scarce. This gap limits a deeper understanding of how perception influences decision-making stability. To address this issue, we propose RoboView-Bias, the first benchmark specifically designed to systematically quantify visual bias in robotic manipulation, following a principle of factor isolation. Leveraging a structured variant-generation framework and a perceptual-fairness validation protocol, we create 2,127 task instances that enable robust measurement of biases induced by individual visual factors and their interactions. Using this benchmark, we systematically evaluate three representative embodied agents across two prevailing paradigms and report three key findings: (i) all agents exhibit significant visual biases, with camera viewpoint being the most critical factor; (ii) agents achieve their highest success rates on highly saturated colors, indicating inherited visual preferences from underlying VLMs; and (iii) visual biases show strong, asymmetric coupling, with viewpoint strongly amplifying color-related bias. Finally, we demonstrate that a mitigation strategy based on a semantic grounding layer substantially reduces visual bias by approximately 54.5% on MOKA. Our results highlight that systematic analysis of visual bias is a prerequisite for developing safe and reliable general-purpose embodied agents. Our code is available at https://anonymous.4open.science/r/Roboview-Bias-CCFD-ee/

## 1 Introduction

The safety and reliability of general-purpose robots depend on accurate and unbiased visual perception, which is the primary channelLiu et al. (2025) through which embodied agentsMa et al. (2024); Li et al. (2024b) perceive and act in the physical world. In hierarchical control, top-level vision-language planners can be biased with respect to color, viewpoint, or scale. Such biases can be amplified as high-level plans are broken into steps and constraints, destabilizing both planning and execution.

Existing robot manipulation benchmarks primarily evaluate an algorithm's generalization James et al. (2020); Zhu et al. (2020); Heo et al. (2023); Pumacay et al. (2024); Luo et al. (2025) and robustness Puig et al.; Xie et al. (2024); Li et al. (2024a) under new tasks and environment changes. However, common metrics emphasize average success rates while overlooking variation and instability across visual attributes, thereby hiding failure risks under specific visual conditions. Specifically, they rarely independently isolate and quantify systematic biases from visual attributes, such as color and camera viewpoint, under controlled conditions. They also lack sensitivity and interaction metrics along the perception-to-decision pipeline, as well as fair and clear comparison sets.

We introduce RoboView-Bias, a benchmark to systematically quantify visual bias in robots using the principle of factorial isolation. To generate evaluation instances, our structured variant-generation framework (SVGF) partitions all variables into two disjoint sets. ❶ Dimensions of Visual Perturbation (V), comprise the attributes under evaluation: 141 object colors, 9 camera orientations, 21 full camera poses, and 9 distance scales. ❷ Dimensions of Task Context Generalization (D), includes 4 initial positions, 4 shapes, and 3 language instructions to ensure robust findings across diverse

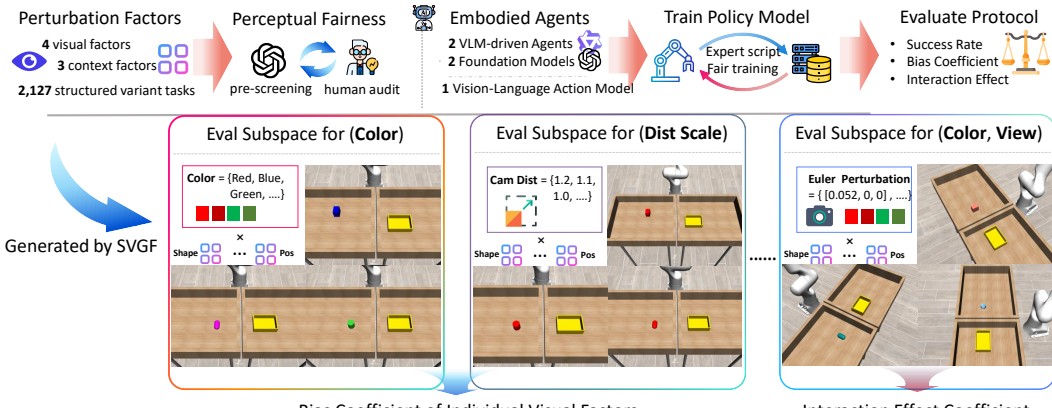

Figure 1: Overview of RoboView-Bias. We construct RoboView-Bias, a benchmark comprising 2,127 task instances, to systematically evaluate visual bias in robotic manipulation. Built upon a factor isolation principle, it enables systematically quantification of how individual visual factors and their interactions impact embodied agent performance and reliability.

task contexts. This methodology yields 2,127 instances and each instance is further validated for perceptual fairness, ensuring it is visually clear and solvable.

In the RoboView-Bias benchmark, we comprehensively evaluated two prevailing paradigms of embodied agents. The results show that these agents exhibit pronounced visual bias. In controlled trials where only the camera viewpoint (pose) varied while all other factors were fixed, success rates fluctuated sharply even across nearby viewpoints, identifying viewpoint as the most influential factor. Similarly, color-focused trials revealed a strong performance bias towards high-saturation hues over achromatic and low-saturation ones, with the extent of the bias varying by agents. In factorial ("color × viewpoint") experiments, analyses of the interaction effect showed that viewpoint changes substantially amplify color-induced performance variation, whereas the reverse effect is weaker. This reveals a strong, asymmetric coupling between the two factors and motivates joint evaluation and mitigation. Based on these observations, we propose the "Semantic Grounding and Perceptual Calibration" (SGL) strategy. We execute pre-training alignment instructions and visible evidence, employing color-invariant calibration to reduce visual bias on MOKA Liu et al. (2024a). This research advances the systematic measurement of visual bias, providing a foundation for bias diagnosis and mitigation to enhance embodied agent stability. Our contributions can be summarized in three key aspects:

- We present RoboView-Bias, a factor-isolated benchmark (color, camera viewpoint) that enables quantitative measurement of visual bias in embodied manipulation.

- We provide cross-paradigm evaluations (VLM-driven, VLA) with fine-grained bias profiles, revealing significant bias and strong, asymmetric color–viewpoint coupling along the perception–decision pipeline.

- We introduce SGL (Semantic Grounding Layer), which aligns commands with visible evidence before execution, reducing visual bias and improving agent stability.

## 2 RELATED WORK

### 2.1 EMBODIED AGENTS FOR ROBOTIC MANIPULATION

Recent advances in Multimodal Large Language Models Achiam et al. (2023); Dosovitskiy et al. (2020), particularly Vision-Language Models (e.g., OpenAI (2024); Bai et al. (2025); Liu et al. (2023); Dai et al. (2023)), and the development of diverse robotics datasets O'Neill et al. (2024); Bu et al. (2025) have inspired two dominant paradigms for instruction following Qin et al. (2024); Wen et al. (2024); Shi et al. (2025) embodied agents. The first involves end-to-end Vision-Language Action Models Driess et al. (2023); Kim et al. (2025); Zitkovich et al. (2023); Black et al., whose

control precision is often limited by action discretization Pearce et al. (2023), leading to recent explorations of diffusion models Chi et al. (2023) as policies or as diffusion decoders Li et al. (2024c); Wen et al. (2025). The second paradigm employs VLMs as high-level planners Huang et al. (2025); Liu et al. (2024a); Zhao et al. (2025); Huang et al. to guide traditional control modules, excelling in zero-shot generalization while their performance is often highly sensitive to implementation details and unbiased collaboration between each submodule. Both paradigms fundamentally rely on the visual perception of the underlying Vision-Language Models, they are vulnerable to inheriting and amplifying latent visual biases. Therefore, we introduce a systematic benchmark to diagnose and quantify these visual biases in embodied agents.

## 2.2 ROBOTIC MANIPULATION BENCHMARKS

The progress in the field of robot manipulation is closely related to the promotion of high-quality benchmarks. Early robotic manipulation benchmarks like RLBench James et al. (2020) and Robosuite Zhu et al. (2020) established standardized evaluation protocols. Subsequent work aimed to assess broader capabilities: benchmarks such as FactorWorld Xie et al. (2024), and THE COLOSSEUM Pumacay et al. (2024) focused on robustness to systematic perturbations, while others like CALVIN Mees et al. (2022) and BEHAVIOR-1K Li et al. (2024a) targeted the challenges of long-horizon tasks. To address the lack of detailed quantitative analysis focused on vision in other benchmarks, we developed RoboView-Bias to assess whether an agent's performance exhibits biases across different visual conditions, enabling a more fine-grained analysis of its perceptual robustness.

## 3 STRUCTURED VARIANT-GENERATION FRAMEWORK

Domain Randomization (DR) Rajeswaran et al. (2017); Pinto et al. (2017); Tan et al. (2018) aims to create a broad training distribution by independently and randomly sampling Brus & De Gruijter (1997); Olken & Rotem (1995) multiple perturbation parameters (such as color, size, and friction) in each iteration. However, its simultaneous sampling of multiple variables is at odds with factorized analysis, making it difficult to disentangle the independent influence of each factor. To enable systematic and attributable bias assessment, we introduce the structured variant-generation framework (SVGF). We reframe scene generation as a programmable generative grammar. A unified interface provides a consistent abstraction layer for all variable factors. Complex generation logic, such as color schemes or grid positions, is then programmatically encapsulated into independent, reusable sampler modules, enabling dynamic, code-level extensibility. A `RecursiveVariantTaskManager` recursively traverses and combines these modules to systematically generate and instantiate task sets.

**Task Selection**. We focus on only one fundamental task, grasping, for the following reasons: First, the vast combinatorial space of variations required for a robust evaluation, even for a single task, presents a substantial yet tractable challenge, making it a suitable starting point for a foundational study. Second, as a canonical manipulation skill, this simple task avoids unfair evaluations caused by some agents being better at specific tasks than others.

**Visual Perturbation Factors**. We adopt three types of visual input perturbations. To conduct color preference analysis, we use 141 named `HTML colors` to perform color perturbation on the robot-manipulated object. These colors are sourced from a recent W3C color name specification. To test viewpoint robustness, we apply 8 minor `camera euler` pose changes to the primary viewpoint, and designed three sets of circular overhead orbit `camera poses`, which are detailed in the Appendix A.1. All viewpoints ensure that key visual information is clearly visible. To introduce `scale` changes, we translate the camera from its initial pose backward along the line-of-sight direction to 8 discrete distance levels, each corresponding to a unique scale factor.

**Task Context Perturbation Factors**. To ensure the evaluation results have better robustness, we perform perturbations by diversifying the task context. We designed 4 `initial positions` for the manipulable object and provided 4 `geometric shapes`. In addition, for the same task goal, we designed 3 types of `task instructions` with identical semantics but different syntax.

**Implementation of Perturbation Factors**. To efficiently implement the dynamic configuration of the aforementioned perturbations, we built our system upon the recently released Roboverse simulation platform Geng et al. (2025). A key advantage of Roboverse is its unified interface that enables

seamless switching between simulators Authors (2024); Coumans & Bai (2016); Makoviychuk et al. (2021); Mittal et al. (2023); Rohmer et al. (2013); Todorov et al. (2012); Xiang et al. (2020), which we leveraged for the initial environment setup. However, for certain dynamic perturbation capabilities not natively supported by Roboverse, such as adjusting object shapes, we implemented them directly using the low-level API of the underlying MuJoCo Todorov et al. (2012) engine.

## 4 PERCEPTUAL FAIRNESS VALIDATION

To ensure the core objective of the RoboView-Bias benchmark, which is to reliably quantify and attribute visual bias, we introduce a rigorous **Perceptual Fairness Validation** pipeline. This process is designed to eliminate confounding variables, such as object occlusion. Our approach contrasts with benchmarks focused on generalization, which may tolerate or even encourage partial observability. To enhance scalability and conserve manual effort, we employ a two-stage validation process combining large-scale automated screening with expert human review.

**Stage 1: VLM-based Automated Pre-screening.** We first leverage GPT-4o as a visual evaluator to screen each generated task instance against a set of predefined clarity criteria detailed in Appendix A.2. We established an iterative refinement loop: if more than 5% of instances are flagged as ambiguous, we manually intervene by adjusting parameters (e.g., object positions) or removing problematic disturbance factors. This cycle is repeated until the pass rate consistently exceeds 95%.

**Stage 2: Human Adjudication**. Following automated screening, all candidate instances undergo a final human review. This stage acts as a crucial quality gate. If the proportion of instances failing this review surpasses a predefined threshold, the entire generation process reverts to Stage 1 for iterative adjustment. This loop continues until a generated batch achieves a pass rate of over 95% in the human adjudication phase, ensuring the high-quality and perceptual fairness of the benchmark.

## 5 EVALUATION PROTOCOL

We propose a evaluation protocol, which first quantifies the performance impact of individual visual factors, then analyzes the interaction effects among those causing significant degradation.

### 5.1 FORMALIZING THE EVALUATION SPACE

All variable factors are partitioned into two mutually exclusive sets.

1. **Visual Perturbation Dimensions** ($V$)**:** This set, $V = \{V_1, V_2, \ldots, V_n\}$, comprises the core visual attributes whose impact we aim to evaluate.

2. **Task Context Dimensions** ($D$)**:** This set, $D = \{D_1, D_2, \ldots, D_m\}$, includes non-visual factors (e.g., $D_{\text{Initial Pose}}$) used to diversify task scenarios.

To ensure that other visual dimensions $V_j$ (for $j \neq i$) remain constant while evaluating a specific dimension $V_i$, we assign a **baseline value** $b_k \in V_k$ for each $V_k \in V$. This value typically represents a standard or common visual setting (e.g., $b_{\text{color}} = \text{red}$). We denote the set of visual baselines by $B$.

### 5.2 THE GENERALIZATION CONTEXT SPACE

The Generalization Context Space ($C_{\text{Gen}}$) is a systematically constructed set of diverse and consistent task scenarios. Each element is a complete, executable task scenario where the value of the dimension under evaluation is left unspecified.

The construction of task configurations, denoted $D_{\text{context}}$, addresses the high computational cost of a full Cartesian product over all dimensions of the task context ($D_1 \times \cdots \times D_m$). We employ a **Structured Union** approach, starting from a baseline configuration $G = (g_1, \ldots, g_m)$ where each $g_k \in D_k$ is a default value. For each dimension $D_k$, we form a **Variation Subspace**, $C_k^{\text{gen}}$, by varying its values while holding all others at baseline.

$$C_k^{\text{gen}} = \{(g_1, \ldots, g_{k-1}, d, g_{k+1}, \ldots, g_m) \mid d \in D_k\} \quad (1)$$

The set of all task configurations is the union of these subspaces, which systematically generates a comprehensive set of scenarios:

$$D_{\text{context}} = \bigcup_{k=1}^{m} C_k^{\text{gen}} \tag{2}$$

To evaluate a specific visual dimension $V_i$, we combine these task configurations with a set of fixed baseline values for all other visual dimensions, $B_{-i} = \{b_j \mid \forall j \in V, j \neq i\}$. The final Generalization Context Space for $V_i$ is then:

$$C_{\text{Gen}}(V_i) = \{d \cup B_{-i} \mid d \in D_{\text{context}}\} \tag{3}$$

This resulting set $C_{\text{Gen}}(V_i)$ serves as the controlled background environment for our bias evaluations.

## 5.3 EVALUATION TASK SUBSPACE

To evaluate a specific visual dimension $V_i$, we define the set of all experimental instances as its **Task Subspace**, $\mathcal{T}(V_i)$. This subspace is formed by the Cartesian product of the values in $V_i$ and the corresponding generalization context space, $\mathcal{C}_{\text{Gen}}(V_i)$:

$$\mathcal{T}(V_i) = V_i \times \mathcal{C}_{\text{Gen}}(V_i) = \{(v, c) \mid v \in V_i, c \in \mathcal{C}_{\text{Gen}}(V_i)\}$$

Each task instance $(v, c) \in \mathcal{T}(V_i)$ is the basis for all subsequent metrics.

## 5.4 METRICS

**Average Success Rate**. The agent's baseline performance is measured by the **Average Success Rate** ($\mu_{SR}$) within a task subspace $\mathcal{T}(V_i)$. It is calculated as the mean of binary success outcomes over all instances.

$$\mu_{SR}(\mathcal{T}(V_i)) = \frac{1}{|\mathcal{T}(V_i)|} \sum_{(v,c) \in \mathcal{T}(V_i)} SR(v, c)$$

**Bias Coefficient**. To quantify performance sensitivity to a visual dimension $V_i$, we introduce the Bias Coefficient ($CV_{SR}(V_i)$). This metric is based on the **Conditional Coefficient of Variation (CCV)** for a fixed context $c \in C_{\text{Gen}}(V_i)$. To improve numerical stability when the mean success rate is close to zero, we add a small bias term $\epsilon$ to the bottom term of the fraction.

$$CV(V_i \mid c) = \frac{\sigma_{v \in V_i}[SR(v, c)]}{\mu_{v \in V_i}[SR(v, c)] + \epsilon} \tag{4}$$

The Bias Coefficient is then the expectation of the CCV over all contexts in $C_{\text{Gen}}(V_i)$.

$$CV_{SR}(V_i) = \mathbb{E}_{c \in C_{\text{Gen}}(V_i)}[CV(V_i \mid c)] = \frac{1}{|C_{\text{Gen}}(V_i)|} \sum_{c \in C_{\text{Gen}}(V_i)} CV(V_i \mid c) \tag{5}$$

**Interaction Effect Coefficient (IEC)**. To capture the coupling between biases, the $IEC(V_i; V_j)$ measures how much the bias from a visual factor $V_i$ is affected by changes in another factor $V_j$.

$$IEC(V_i; V_j) = \mathbb{E}_{c \in C_{\text{Gen}}(V_i, V_j)} \left[ \frac{\sigma_{v_j \in V_j}[CV(V_i \mid v_j, c)]}{\mu_{v_j \in V_j}[CV(V_i \mid v_j, c)]} \right] \tag{6}$$

# 6 EXPERIMENTS AND EVALUATION RESULTS

## 6.1 BASELINES

**VLM-driven Embodied Agents.** ❶ The first agent we evaluate is SimpleAgent (hereafter referred to as Simple), a minimalist embodied agent based on the embodied LLM prototype introduced in BadRobot Zhang et al. (2025). It consists of a single VLM coupled with a heuristic action policy. By design, this agent intentionally omits specialized perception grounding modules. This minimalist structure allows us to directly expose the inherent visual perception biases of the VLM when confronted with physical world tasks—biases that may be a potential source of error in more complex VLM-driven agents. ❷ The second agent, MOKA Liu et al. (2024a), connects a VLM's 2D image

| Embodied Agents | Color | | Camera Pose | | Camera Euler | | Dist Scale | | Average | |
|---|---|---|---|---|---|---|---|---|---|---|
| | SR | CV | SR | CV | SR | CV | SR | CV | SR | CV |
| MOKA(Qwen-VL-Max) | 22.92 | 139.25 | 38.10 | 91.68 | 56.16 | 49.2 | 68.89 | 35.16 | 46.52 | 78.82 |
| MOKA(GPT-4o) | 23.92 | 134.54 | 68.23 | **40.28** | 71.72 | **28.38** | 70.51 | **28.67** | 58.60 | 57.97 |
| Simple(Qwen-VL-Max) | 47.83 | 107.25 | 38.10 | 96.4 | 12.93 | 197.23 | 34.55 | 92.61 | 33.35 | 123.37 |
| Simple(GPT-4o) | 23.00 | 137.23 | 1.56 | 175.11 | 0.00 | N/A | 1.41 | 178.12 | 6.49 | N/A |
| $\pi_0$ | 53.87 | **37.63** | 30.22 | 84.87 | 57.78 | 36.81 | 44.24 | 52.90 | 46.53 | **53.05** |

Table 1: Performance Evaluation of Embodied Agents on Visual Perturbation Dimensions. The table reports the Average Success Rate (SR), corresponding to $\mu_{SR}$, and the Bias Coefficient (CV), corresponding to $CV_{SR}$. Bold values indicate the best performance in each CV column.

predictions to 3D robot actions. It leverages advanced grounding models (e.g., Grounding DINO Liu et al. (2024b), SAM Kirillov et al. (2023)) and mark-based visual prompting to generate compact, point-based affordance representations. MOKA is designed to solve open-world manipulation tasks from free-form language instructions in a zero-shot manner. We replicated MOKA in simulation, where it performed our tasks effectively after merely adjusting its configuration parameters.

**Vision-Language Action Models.** The VLA model, $\pi 0$, is built on a flow matching Lipman et al. (2022) architecture, a variant of diffusion models, to effectively model complex, continuous action distributions. It uses a pre-trained Vision Language Model (PaliGemma Beyer et al. (2024)) as its backbone and is trained on over 10,000 hours of cross-embodiment data. The model exhibits strong out-of-the-box performance and instruction-following capabilities. RoboView-Bias employs an expert algorithm to collect demonstration data via standardized script, ensuring training fairness. The collected data includes *rgb and depth* from four camera views (wrist, front, left, and right). During data collection, we apply domain randomization *exclusively* to Task Context Perturbation factors. Detailed configurations are available in the Appendix A.3.

### 6.2 EXPERIMENTAL SETUP

For each embodied agent, we first measure its Bias Coefficient for every visual perturbation across the generalization context space $C_{\text{Gen}}$. Each specific task instance for this analysis is run **5 times**.We then focus on two specific visual dimensions, color and camera pose, to measure their Interaction Effect Coefficient (IEC). Due to computational constraints, this IEC analysis is not performed on the entire $C_{\text{Gen}}$ space. Instead, the evaluation is conducted within a fixed, representative context ($c^*$) using a default parameter configuration. and each task instance is run **10 times**. For MOKA and SimpleAgent, if not specifically labeled, the basic model uses Qwen-VL-Max.

### 6.3 INDIVIDUAL VISUAL BIAS

**VLM-driven embodied agents** commonly exhibit significant visual bias. When based on the `Qwen-VL-Max` model, the mean visual biases (CV) of MOKA and Simple are as high as 78.82% and 123.37%, respectively (see Table 1). As a minimalist prototype, SimpleAgent not only has the highest average visual bias but is also extremely sensitive to camera euler changes, with its bias coefficient surging to 197.23% after a slight adjustment in camera angles. In contrast, by integrating modules for grounded perception and low-level control, MOKA significantly reduces its overall visual bias, achieving a CV score more than 40 points lower than that of SimpleAgent. Notably, its bias remains extremely high in the color dimension, which can likely be attributed to error accumulation within its multi-module architecture. The choice of VLM critically impacts this bias: MOKA shows lower overall bias with `GPT-4o` compared to its `Qwen-VL-Max` version.

**The VLA model**, $\pi_0$, displays relatively balanced overall stability but still possesses a visual bias of 53.05%. The robustness of $\pi_0$ to color variations is far superior to that of the VLM-driven agents, with a bias rate of only 37.63%. It also maintains a low bias (36.81%) and a high success rate (57.78%) under slight perturbations of the camera's euler angles. However, when the entire camera pose undergoes drastic changes, its bias rate significantly increases to 84.87%, revealing its limited generalization capability in spatial visual perception.

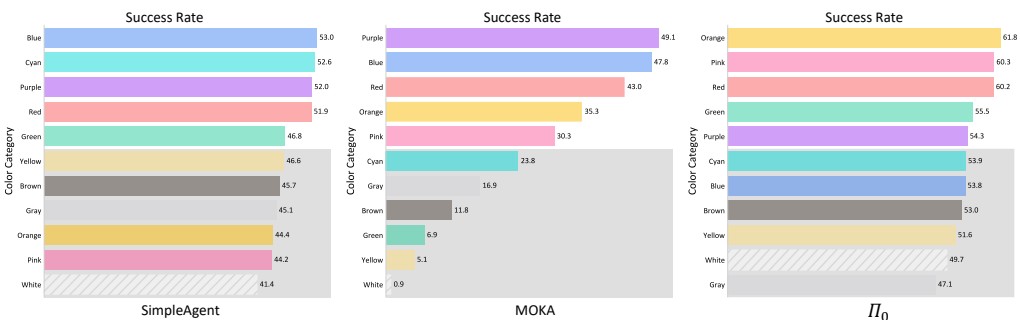

Figure 2: Average success rate for each embodied agent, grouped by color category. The rates are calculated over the entire color task subspace ($\mathcal{T}(V_{\text{color}})$).

**In the color dimension**, as shown in Figure 2, our analysis reveals a systematic color perception bias common to all evaluated agents. First, all agents demonstrate consistently lower success rates for achromatic or low-saturation colors, such as gray and white. In contrast, their performance is generally higher when handling salient, high-saturation colors like red. This finding indicates that the performance of current embodied agents relies heavily on salient color features, a general bias likely inherited from their underlying vision foundation models.

**In the camera pose dimension**, all agents are highly sensitive to changes in camera pose. As illustrated in Figure 3, their success rates fluctuate sharply with variations in camera pose. A key finding is that all agents have specific viewpoints that lead to complete task failure. Furthermore, they can also achieve higher success rates from perturbed viewpoints compared to their original poses. This phenomenon clearly indicates that the performance of current agents is tightly coupled with their observation perspective. This also provides a potential direction for future research: developing algorithms that can find the optimal viewing perspective or equip agents with active vision capabilities is of critical importance for enhancing their overall robustness and performance.

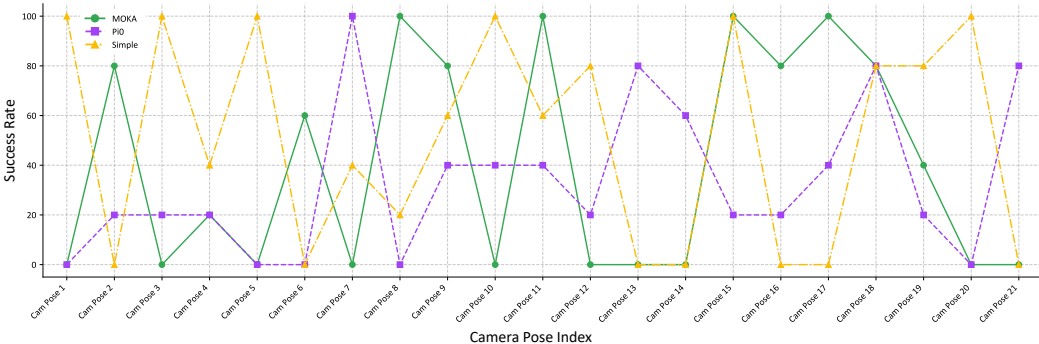

Figure 3: Success Rate of MOKA, $\pi_0$, and SimpleAgent under various camera pose perturbations. The evaluation is conducted within a specific context from the task subspace $\mathcal{T}(V_{\text{camera pose}})$.

## 6.4 INTERACTION EFFECTS OF COLOR AND CAMERA POSE

As illustrated in Figure 4 and quantified in Table 2, our evaluation reveals a significant **asymmetric dependency** between camera pose and color. The heatmaps visually suggest this imbalance, showing that performance patterns are often more distinctly stratified by camera pose (rows) than by color (columns). This observation is numerically confirmed by the data: on average, the bias from camera pose ($CV_{SR}(P) = 125.25$) is substantially higher than from color ($CV_{SR}(C) = 113.93$). Furthermore, the interaction is lopsided, as the influence of pose on color bias ($IEC(C; P) = 57.06$) is nearly double the reverse effect ($IEC(P; C) = 29.50$). The agents show a tendency to be more sensitive to variations in camera pose than in color, which further highlights their limited 3D spatial perception. However, specific agents like MOKA exhibit a mutual dependency between these two

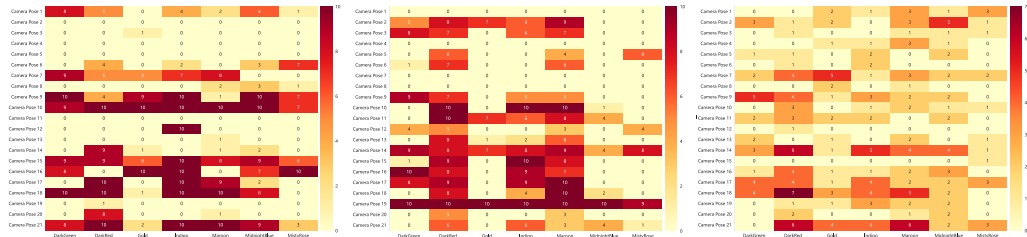

Figure 4: Heatmaps of success counts for the Simple (left), MOKA (middle), and $\pi_o$ (right) agents. Each cell represents the performance for a unique combination of camera pose (row) and object color (column), visualizing the interaction between these two visual dimensions.

| Embodied Agents | $CV_{SR}(C)$ | $IEC(C;P)$ | $CV_{SR}(P)$ | $IEC(P;C)$ |
|---|---|---|---|---|
| MOKA | **100.11** | **42.39** | 138.83 | 50.96 |
| Simple | 132.17 | 70.48 | 132.64 | **18.17** |
| $\pi_0$ | 109.52 | 58.32 | **104.27** | 19.37 |
| Avg | 113.93 | 57.06 | 125.25 | 29.50 |

Table 2: Evaluation in a task space with changing color (C) and camera pose (P). $CV_{SR}(V)$ is the performance bias from factor $V$. $IEC(V_i;V_j)$ measures how much the bias from $V_i$ is affected by changes in $V_j$. Lower values are better. **Bold** is best, underlining is worst.

visual factors ($IEC(C;P) = 42.39$ and $IEC(P;C) = 50.96$). This finding highlights the necessity of analyzing their interaction effects to develop targeted improvements for different agents.

## 6.5 CASE STUDY: ANALYSIS OF THE MOST COLOR-BIASED EMBODIED AGENT

Of the three embodied agents we evaluated, MOKA exhibited the most significant color bias. To investigate its root cause, we analyzed two stages of its workflow.

❶ During the high-level planning stage in MOKA, the VLM responsible for task decomposition exhibits significant descriptive preferences. It generates inconsistent descriptions for identical objects—for instance, describing the same block as "geometric object," a "block," or a "red block" (details in Appendix A.4). This descriptive inconsistency, particularly the arbitrary omission or inclusion of color attributes, directly impacts the performance of downstream modules. As shown in Figure 5, the most frequent colors in the VLM's descriptions are gray, red, blue, and green. While the prevalence of gray may be due to misclassification from object shadows, we speculate that red, green, and blue appear frequently because, they are among the most common colors.

❷ A perceptual deviation exists between the color understanding of the VLM and the perception of Grounding DINO during the visual grounding stage. To quantify this, we conducted an experiment where we replaced the original color descriptions of the VLM with similar colors from the color space to create new labels. A significant perceptual deviation was confirmed if the localization confidence score of the new label was substantially higher (threshold = 0.03) than that of the original. The results (Figure 6) show this occurred in 17.78% of cases, confirming a significant perceptual difference between the two modules.

In summary, the severe color bias ultimately exhibited by the system stems from the compounding and cumulative amplification of semantic bias at the planning stage and visual bias at the perception stage. Therefore, for complex modular embodied systems like MOKA, eliminating such internal biases between modules and ensuring alignment from high-level semantics to low-level vision is the core premise for achieving robust generalization in the open world.

## 6.6 MITIGATING BIAS VIA SEMANTIC GROUNDING: A PROPOSED APPROACH

While standardized instructions are intuitive for humans, they can be semantically ambiguous for embodied agents. In the MOKA system, we identify this ambiguity as a key source of perfor-

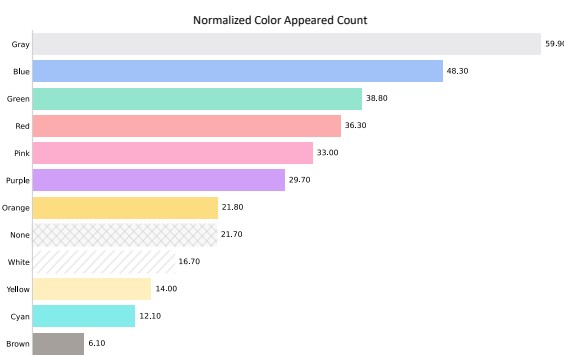

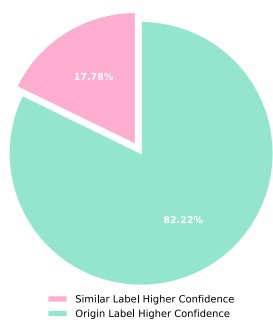

Figure 5: Normalized count of colors appearing in the subtask descriptions generated by the VLM (qwen-vl) in MOKA during the high-level planning stage.

Figure 6: Quantifying the perceptual deviation between the VLM (qwen-vl) and Grounding DINO.

mance bias, a problem often overlooked in robotics. Such ambiguity can degrade the performance of downstream policy models by creating uncertainty in task execution. To address this, we propose a **Semantic Grounding Layer (SGL)**. The core idea is to resolve semantic ambiguity by grounding the language instruction in its visual context before execution. The SGL operates in three stages:

❶ **Scene Parsing and Action Decomposition:** Given an instruction $I_{orig}$ and a visual scene $V$, a VLM first identifies all relevant objects and their attributes while extracting the core action.

❷ **Ambiguity Detection and Attribute Selection:** To perform perceptual calibration, the layer uses heuristic rules to detect potential ambiguities across various dimensions (details in Appendix A.5).

❸ **Instruction Refinement:** Finally, the SGL synthesizes a refined instruction by combining the action with the selected attributes. For instance, an ambiguous instruction like "stack the cube" is transformed into the clear, executable command "put the small red cube on the larger cube."

To validate our approach, we integrated the SGL into each evaluated agent and re-assessed their performance on our bias benchmarks, using both object color and the task instruction as perturbation factors. As shown in Figure 7, SGL mitigated the visual bias in MOKA by 54.5%. The improvements for SimpleAgent and $\pi_0$ were less pronounced. We attribute this to the simplistic and monolithic nature of the current task scenarios, and the method's efficacy in complex environments requires further study.

## 7 CONCLUSION AND FUTURE WORK

This paper introduces RoboView-Bias, the first benchmark for systematically quantifying visual bias in embodied manipulation agents. By constructing a highly structured benchmark and comprehensively evaluating agents from the two dominant paradigms, we reveal pervasive visual biases, especially a strong sensitivity to camera pose and coupling effects among different visual factors. Finally, based on an in-depth analysis of the sources of bias, we propose a Semantic Anchoring Layer as a potential method for mitigating visual bias. We hope this work will encourage further research into the visual perception stability of embodied agents.

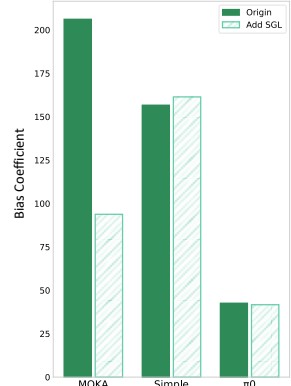

Figure 7: Comparison of the Bias Coefficient for each agent before (Origin) and after integrating the Semantic Grounding Layer (SGL).

**Limitations.** Despite our best efforts, we acknowledge several limitations and would like to explore the following directions in future work: first, expanding the benchmark's scope to include more diverse visual factors (e.g., material properties, lighting) and manipulation tasks (e.g., pushing); second, evaluating a broader and more architecturally diverse set of VLA models to understand the influence of architecture on bias; third, investigating the sim-to-real gap for bias assessment by correlating simulated findings with real-world performance.

## 8 ETHICAL CONSIDERATIONS

Our research aims to identify and quantify visual biases in embodied agents, a critical step toward ensuring the safety, fairness, and reliability of future robotic systems. All of our experiments and evaluations are conducted within a fully controlled simulation environment (Roboverse). This approach allows us to systematically analyze and diagnose biases that could lead to failure, without posing any physical risk to people or property in the real world. Crucially, we not only identify the problem but also propose and validate a mitigation strategy, the Semantic Grounding Layer (SGL), to address the potential negative impacts of these biases. We believe this work contributes to the development of more robust and trustworthy general-purpose robots and encourages the community to focus on and address potential biases in AI systems.

## 9 REPRODUCIBILITY STATEMENT

To ensure the reproducibility of our work, we have provided the source code, which is available at an anonymized link. Our evaluation protocol is defined in detail in **Section 5**, which includes the complete task space setup and metric design. To guarantee the validity of our evaluation, we designed and implemented a rigorous Perceptual Fairness Validation pipeline (**Section 4**) and have provided the full prompts used for automated screening in **Appendix A.2**. Furthermore, we introduce the architecture and principles of our proposed bias mitigation method, the Semantic Grounding Layer (SGL), in **Section 6.6**, with its specific implementation details further elaborated in **Appendix A.5**.

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

## A APPENDIX

### A.1 CAMERA SETTINGS

We deployed a diverse set of cameras within the simulation environment, as illustrated in Figure 8. Our camera setup includes:

1. **Four manually positioned cameras:** These provide a broad view of the workspace and robot arm from different angles: left top, right top, front, and a wrist camera.

2. **Twenty-one orbital cameras:** Three sets of seven cameras are arranged in concentric rings, providing a top-down, panoramic view in front of the robot arm.

3. **Nine camera poses (Euler angles):** In addition to the original camera, we introduced eight minor disturbances to the Euler angles of the manually positioned front and left top cameras. Specifically, we applied these eight variations by rotating the yaw and pitch from -6° to 6°, resulting in nine distinct perspectives.

4. **Nine camera positions (Translations):** In addition to the original camera position, the positions of the front and left top cameras were shifted eight times, in increments of 0.05 units, to simulate disturbances of the dist scale. This resulted in a total of nine distinct positions.

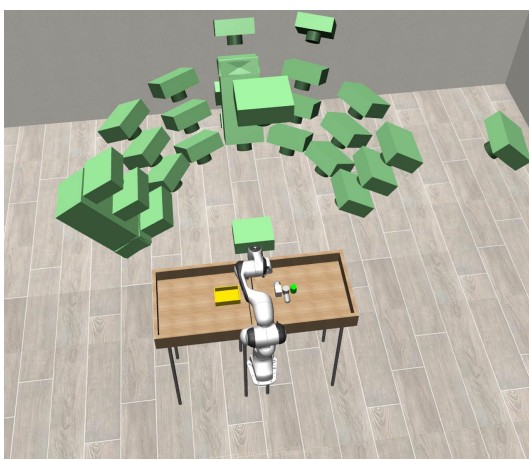

Figure 8: The camera setup in the simulation environment.

### A.2 PROMPTS USED TO VALIDATE PERCEPTUAL FAIRNESS AND EXAMPLE RESULTS

Our automated pre-screening leverages a Visual-Language Model (VLM), specifically GPT-4o, to act as a visual evaluator. The model's primary task is to assess each generated image for clarity and adherence to specific criteria. This process helps us filter out ambiguous or confusing instances at a large scale before human review. The prompt template (Figure 9) is used to guide the VLM's evaluation. It is designed to be highly specific to minimize misinterpretations and ensure a consistent assessment.

```
You are an expert image analyst acting as aJSON API. Your task is
    to analyze the provided image and return a single, valid JSON
    object. Do not output any text before or after the JSON object
    .
**1. Conditions to Check:**
- Is there a small solid geometry object in the image?
- Is there a yellow box in the image?
- Can a person clearly identify both the geometry object and the
    yellow box?

**2. Rule for the Final Answer:**
- If ALL THREE conditions are true, the `final_answer` value is "
    yes".
- If ANY condition is false, the `final_answer` value is "no".

**3. Required JSON Output Format:**
Your entire output must be a single JSON object with these two
    keys:
- `analysis`: A string containing a brief explanation of your
    reasoning.
```

```
- `final_answer`: A string that is either "yes" or "no".
---
**Example 1 (All conditions met):**
{
  "analysis": "The image clearly shows a small blue pyramid and a
      yellow box, and both are identifiable.",
  "final_answer": "yes"
}
**Example 2 (One condition fails):**
{
  "analysis": "The image contains a small pyramid, but the box is
      red, not yellow.",
  "final_answer": "no"
}
---
Now, analyze the image I provide and respond only with a valid
    JSON object as specified.
```

Figure 9: Prompts used to validate perceptual fairness through GPT-4o.

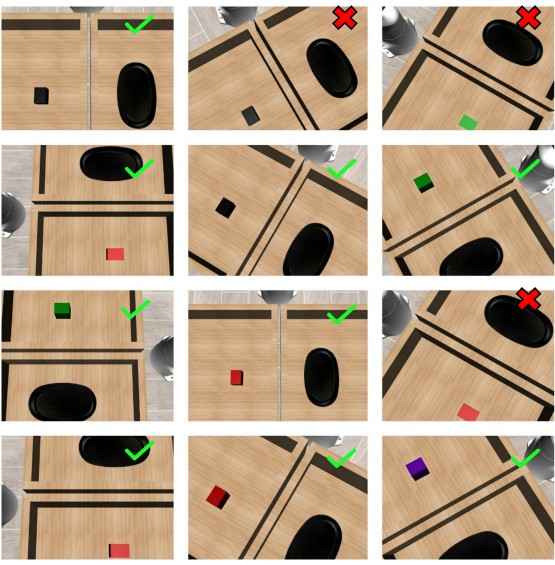

Figure 10: Perceptual Fairness Validation Results (Case 1) Using GPT-4o.

Figures 10, 11, and 12 illustrate several examples of successful and failed evaluation outcomes. Initially, certain camera viewpoints were unevaluable because they failed to capture the three-dimensional nature of the blocks, making them appear as flat 2D shapes. This perceptual ambiguity made a definitive evaluation impossible. In such cases, we iteratively adjusted the viewpoints manually until a definitive evaluation was possible.

### A.3 TRAINING DETAILS

To generate our training data, we first create task instances by applying domain randomization over the task context perturbation factors. We then leverage a standard script to collect a total of 350 demonstration trajectories. We fine-tune the publicly available $\pi_0$-droid checkpoint released by openpi. The model takes RGB images from two manually configured camera views as input: a gripper camera and a top-left camera. The entire fine-tuning process was conducted on a single NVIDIA A100 GPU for 10,000 iterations with a batch size of 16. We employed a cosine annealing learning rate schedule, where the learning rate decayed from an initial value of $5 \times 10^{-5}$ to a final value of $2.5 \times 10^{-5}$.

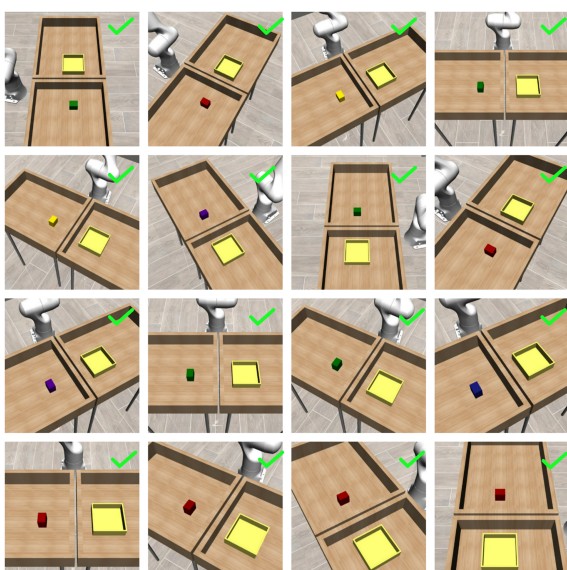

Figure 11: Perceptual Fairness Validation Results (Case 2) Using GPT-4o.

Figure 12: Perceptual Fairness Validation Results (Case 3) Using GPT-4o.

### A.4 SHAPE DESCRIPTOR BIAS IN MOKA'S HIGH-LEVEL PLANNING

We analyzed the shape descriptors generated by the VLM that feeds into MOKA's high-level planner and found a significant vocabulary imbalance, as shown in Figure 13. The model heavily favors a few common terms, with cube (30.0%), cylinder (23.9%), and the generic word object (22.4%) collectively comprising over 75% of its vocabulary. This pattern indicates that the VLM simplifies diverse geometries into a few familiar categories—a bias likely inherited from its training data that directly affects downstream planning.

### A.5 IMPLEMENTATION DETAILS OF SEMANTIC GROUNDING LAYER

In the parsing stage of the SGL, we first considered the characteristics of our experimental environment. As the visual scenes are relatively simple and controlled, we found that we could achieve effective and stable scene parsing by simply designing a structured prompt to guide the VLM. The

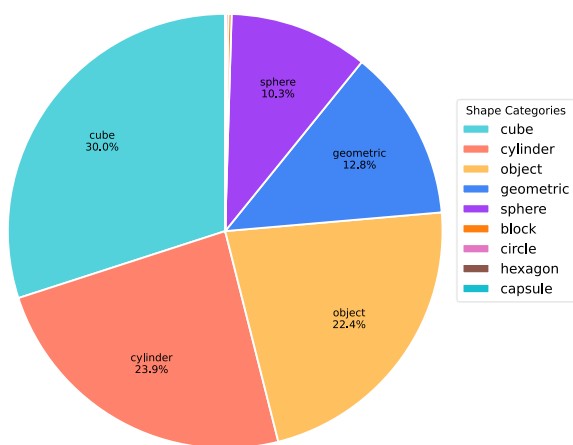

Figure 13: The frequency distribution of shape descriptors generated by the VLM during MOKA's planning phase.

core of this prompt is shown in Figure 14, is to leverage human prior knowledge to instruct the VLM. It directs the model to identify key objects and extract a set of attributes, including their category, color, size, position, and physical state. This direct approach has proven sufficient for the scope of our current tasks. To enable generalization to more diverse environments in the future, we suggest constructing a library of perceptual priors for different scene types, which would allow the SGL to adapt its parsing strategy dynamically. Following the initial scene parsing by the VLM, the SGL performs ambiguity detection and attribute selection using a set of simple heuristic rules. This approach is particularly effective for our current, controlled scenes. The process begins by identifying a potential ambiguity, which occurs when multiple objects share a common category (e.g., `geometry`). To resolve the ambiguity, the system evaluates object attributes based on a fixed priority (color > state > size > position) and selects the most discriminating attribute value to use as a prefix. This generates a precise description, such as "left red cube".

```
You are an expert vision assistant for a robot. Your task is to
    analyze a visual scene and a user instruction to identify all
    relevant objects and their properties. Your final output must
    be a single, valid JSON list.

The user's instruction is: "{instruction}"

---
### **1. Object Identification Rules**
Based on the instruction and the scene, you must identify:
- **One 'manipulation object'**: The primary object to be moved or
    interacted with.
- **Zero or one 'receiver object'**: The object that receives the
    manipulation object (e.g., a box, a table).
- **'n' other objects**: Any other clearly visible objects in the
    scene.

### **2. Required Object Attributes**
Each object in the output list must have the following attributes:

- **"ID"**: A unique integer identifier for the object.
- **"object_type"**: A string, must be one of: ''manipulation
    object'', ''receiver object'', or ''other object''.
- **"name"**: A short, essential noun for the object (e.g., 'box',
    'cube', 'pyramid').
- **"category"**: A list of common categories. You must carefully
    consider shared properties. For example:
```

```
        – A cube and a pyramid are both ''geometry''.
        – A cube and a box can both be ''rectangular shape''.
        – If multiple objects share a category, you MUST include that
            shared category for all of them.
    – **"color"**: The object's color. For ambiguous colors, combine
        the names (e.g., "purple blue", "gray white").
    – **"size"**: A string, must be one of: ''small'', ''normal'', or
        ''big'', judged relative to other objects in the scene.
    – **"position"**: The object's location if obvious (''left'', ''
        right'', ''top'', ''bottom''). Otherwise, use ''normal''.
    – **"state"**: The object's physical structure, must be one of: ''
        solid'' or ''hollow''.

    ---
    ### **3. Example**
    **Instruction:** "Put the small geometry into the box"
    **Scene:** A small, solid red cube on the left; a normal, hollow
        yellow box on the right; and a normal, solid blue pyramid in
        the middle.

    **Expected Output (Format Reference Only):**
    [
        {
            "ID": 1,
            "object_type": "manipulation object",
            "name": "cube",
            "category": ["cube", "geometry", "rectangular shape"],
            "state": "solid",
            "color": "red",
            "size": "small",
            "position": "left"
        },
        {
            "ID": 2,
            "object_type": "receiver object",
            "name": "box",
            "category": ["box", "container", "rectangular shape"],
            "state": "hollow",
            "color": "yellow",
            "size": "normal",
            "position": "right"
        },
        {
            "ID": 3,
            "object_type": "other object",
            "name": "pyramid",
            "category": ["pyramid", "geometry"],
            "state": "solid",
            "color": "blue",
            "size": "normal",
            "position": "middle"
        }
    ]
    ---
    **Note:** You must only refer to the **format** of the example
        output. The content of your response must be based on the
        actual image and instruction provided to you.
```

Figure 14: Prompts for analyzing scenes based on prior knowledge of human scenarios.

## B    THE USE OF LARGE LANGUAGE MODELS

As part of our commitment to producing a clear and well-written manuscript, we utilized a large language model (LLM) to refine and polish portions of the narrative. The LLM's role was strictly limited to improving the language and readability of our existing text. All scientific claims, experimental designs, results, and conclusions were conceived and articulated by the authors.

