# OpenReview forum: "RoboView-Bias: Benchmarking Visual Bias in Embodied Agents for Robotic Manipulation"
_ICLR.cc/2026/Conference — Submitted to ICLR 2026_

### Official Review · Reviewer_RXBV · 2025-10-29

**Soundness:** 2
**Presentation:** 1
**Contribution:** 2
**Rating:** 2
**Confidence:** 4

**Summary:**

The paper introduces RoboView-Bias, a benchmark for analyzing visual bias in embodied agents built on foundation models. It isolates factors such as color, camera pose, and distance to evaluate how visual variation affects task performance. The authors propose metrics to quantify bias and show that color and viewpoint strongly influence results. They evaluate three embodied agents, two VLM-driven and one VLA model, and find that all exhibit strong visual biases, particularly sensitivity to camera viewpoint and color saturation. A case study on the MOKA agent identifies bias arising from mismatched language and vision modules. They address this by refining ambiguous instructions and reducing color bias.

**Strengths:**

1. The motivation is clear. The paper tackles visual bias in (foundation-model-based) embodied agents, distinct from the more common notion of “robustness under perturbation”. As a result, I am sufficiently convinced that analyzing behavioural bias from pre-training data, such as camera angles, distances, and colors, matters for reliability in embodied perception, apart from just robustness testing.
2. Benchmark design. The factor isolation principle is well thought out. Varying one visual factor at a time while holding others constant gives interpretable results. The structured variant-generation framework (SVGF) is more principled than the random domain randomization used in prior works.
3. Explicit bias metrics. The paper introduces additional metrics (μₛᵣ, CVₛᵣ, IEC) to measure both individual and interaction effects. Despite the overcomplicated notation, the idea is meaningful: measure not just mean performance, but stability under visual variations.
4. Findings. The asymmetry between color and viewpoint bias is interesting. Also, the insight that viewpoint bias amplifies color bias (but not vice versa) is credible and well interpretable.
5. The case study on MOKA’s color bias is a useful point. It goes beyond descriptive benchmarking by identifying how semantic inconsistencies in the planner (VLM) and perceptual deviations in the grounding module (DINO) compound to produce systematic color bias.

**Weaknesses:**

1. The paper is very difficult to follow due to heavy LLM use for writing. The text is saturated with tautologies and filler phrasing such as *“we systematically quantify,” “we comprehensively evaluated,” “principle of factor isolation,”* and *“structured variant-generation framework.”* Such repeated, formulaic expressions rather inflate simple statements than clarify them. The writing also relies heavily on generic claim templates like *“systematically quantify visual bias”* and *“robust measurement of bias,”* which are reiterated with minor wording changes (*“systematic measurement,” “factor isolation principle,” “robust generalization,” “a rigorous Perceptual Fairness Validation pipeline”*). Such circular phrasing gives the appearance of methodological depth without adding substantive explanation. Additionally, there is inconsistency in terminology: the SGL method is called *“Semantic Grounding and Perceptual Calibration,” “Semantic Grounding Layer,”* and then *“Semantic Anchoring Layer.”* Pick one and stick to it. Finally, some sentences, e.g., *“We execute pre-training alignment instructions and visible evidence”,* read as syntactically correct but semantically incoherent. This non-exhaustive list of examples is a hallmark of LLM-generated text. One of the goals of a research paper is to clearly convey its ideas to the readers, and this work falls short on that front.
2. Section 5 tries too hard to sound formal, which makes the notation look heavier than it needs to be. It’s written in a way rather to impress than help readers understand.
    1. Sets and tuples are mixed: G = (g₁, …, gₘ) is treated as a tuple, but then union subspaces C_genₖ that are sets of tuples. It’s better to explicitly say each C_genₖ ⊂ D₁ × … × Dₘ.
    2. Equations (1) and (2) define  𝐶_genᵏ and 𝐷_context overengineer the notation for “we vary one context dimension at a time while keeping others fixed.” Writing two equations for this adds nothing but pseudo-rigor.
    3. The introduction of both CV(Vᵢ∣c) and CVₛᵣ(Vᵢ) is redundant. CVₛᵣ(Vᵢ) is merely the averaged coefficient of variation, yet it’s treated as a distinct “Bias Coefficient.” Readers have to mentally juggle CV, CVₛᵣ, CCV, all defined within a few lines.
    4. The authors define 𝑇(Vᵢ) = Vᵢ × 𝐶_gen(Vᵢ) = { (v, c) | v ∈ Vᵢ, c ∈ 𝐶_gen(Vᵢ) }. This is just a Cartesian product between visual values and context configs. The notation looks formal but contributes no new insight. The same meaning could be stated in English without math symbols. Also, “Task Subspace” is a misleading name. It’s not a subspace in any vector-space sense, just a subset.
    5. The Generalization Context Space is defined as 𝐶_gen(Vᵢ). Later, in Eq. 6, the expectation over “context” uses 𝐶_gen(Vᵢ, Vⱼ) without prior definition. The reader has to guess that it means the context space for both variables varied together. A single sentence and a simple correlation metric would have communicated this more clearly.
    6. Within almost a single-page span, the authors define 𝐷_context, 𝐶_gen, 𝐶_genᵏ, 𝑇(Vᵢ), 𝐵, 𝐵⁻ⁱ, μₛᵣ, CVₛᵣ, and IEC. All for what amounts to “vary one thing, keep others fixed, measure mean and variance.”
3. The “multi-stage fairness validation” is weaker than claimed. It relies on a VLM to flag inconsistent renderings, inheriting the VLM’s own perceptual biases. The human reviewers only validate a small subset of cases. A more reliable solution would be to implement programmatic visibility checks. Segmentation and depth maps could be used to verify that all target objects are fully visible, unoccluded, and correctly colored.
4. Although the paper reports 2,127 valid environments, this figure is largely inflated by fine-grained color enumeration within a single underlying manipulation task. The benchmark covers only one type of task, and while camera pose and distance variations contribute some genuine diversity, most of the 141 color variants occupy nearly identical regions in perceptual color space. Moreover, the evaluation does not explore the full Cartesian product of factors: each dimension is varied independently, resulting in many near-duplicate scenes. Figure 2 already bins the 141 colors into 11 groups, which appears far more realistic, as only those broader distinctions meaningfully affect model behavior. Nowhere in the results is there evidence that fine-grained color distinctions, such as between magenta, violet, and orchid, lead to any measurable difference in performance. These hues all effectively map to a “purple-like” region for the models. With roughly 11 effective colors and correlated camera–distance settings, the number of genuinely distinct and valid environments would likely shrink to only a few dozen. In effect, the task instance diversity is much narrower than claimed.
5. The evaluation includes only three agents, two of which are variants of the same VLM-driven setup and share substantial perception components. Even the supposedly distinct paradigms (VLM-driven vs VLA) depend on similar pretrained visual encoders. As a result, the claim of discovering visual biases across paradigms is not as general as claimed. It’s unclear whether the observed patterns stem from architecture-level biases or simply from shared visual foundations.
6. The reported color perception bias is not as “*systematic*” as the authors claim. While Orange and Pink result in some of the lowest performance for SimpleAgent, they yield the highest success rates for π₀.
7. The proposed SGL is an interesting attempt to reduce ambiguity between language and perception, but it comes across as somewhat artificial. It relies on privileged scene information to detect ambiguities and rewrite instructions, effectively giving the system access to ground-truth object attributes. Even setting that aside, the approach is not very generalizable. It depends on hand-crafted rules and requires manual effort to define relevant attributes and disambiguation logic. As such, it fits neatly into the controlled cube-stacking setup used in this paper but would not extend naturally to other embodied tasks or less modular architectures.

### Minor points

1. The title of Section 5.3 is glued to the text above it, with no spacing left in between
2. The CVₛᵣ metric assumes SR distributions are unimodal and roughly continuous, which is shaky for discrete categories (like colors).
3. The IEC metric can be very noisy in practice. Compounding variances on variances amplifies randomness. Without many samples, it’s statistically fragile.
4. Vertical lines in Table 1
5. The text in Figures 2 and 4 is difficult to read.
6. In Figure 3, the dots are connected by lines, which visually suggests a continuous relationship between discrete visual categories (camera poses). Since these factors are categorical, connecting the points makes little sense. They should be plotted as unconnected markers to avoid implying continuity.

### Typos

1. Line 36 channelLiu et al & agentsMa et al
2. Line 70 enables systematically quantification → enables systematic quantification
3. Line 189 a —> an
4. Line 301 5 times.We
5. Line 305 configuration. and each —> configuration, and each / configuration. Each
6. line 386 πo —> π0

I am happy to raise my score if the points above are addressed. However, in its current form, the paper is not yet up to par with the standards of a top-tier venue like ICLR.

**Questions:**

1. Why are the colors reported in Figure 4 different than the ones used in Figure 2?
2. Instead of only measuring variance, why not quantify directional bias (e.g., which color is favored)? If, say, SimpleAgent and MOKA both prefer red, but π₀ prefers blue, their bias directions are not aligned. This could be done, for instance, by computing signed deviations from the overall mean and ranking these differences. Figure 2 already qualitatively shows these trends, but it would be interesting to quantify this.
3. Why are the bottom half of the colors in Figure 2 in a gray box?
4. How could the SGL approach be adapted to other settings or methods?

---

### Official Review · Reviewer_dtBd · 2025-10-31

**Soundness:** 2
**Presentation:** 3
**Contribution:** 2
**Rating:** 6
**Confidence:** 3

**Summary:**

This work introduces a benchmark to measure visual biases in visual language models used for robotics. The paper particularly focuses on biases arising due to colors of the objects and camera poses used to capture environmental observations. They first create a combinatorial framework to vary individual components and measure the impact of change in a certain variable on the success rate of a task. For measurement of the bias, the paper uses conditional coefficient of variation and to measure the coupling between two different biases it measures interaction effect. The experimental analysis involves 3 different agents. The main results show that MOKA agent has low camera pose bias, but high color bias with both Qwen-VL and GPT-4o backbones. Further, the authors conduct a case study on the color bias of MOKA agent where they attribute this bias to biases in the underlying visual models. To ameliorate these biases, the paper proposes semantic grounding where they detect ambiguity in the scene description and modify the attributes that would lead to poor performance. Later it is shown that this approach reduces the coefficient of variation in MOKA by 54.5%.

**Strengths:**

1) The paper shows that there is significant bias in visual language models with respect to colors of entities. This is insightful as the models are usually considered powerful enough to be robust against innocuous changes in colors.
2) The methodology is straightforward and allows to remove various confounding factors in the analysis. I especially like the choice of individual modification the component and performing analysis on the single manipulation task.
3) The semantic grounding feels promising given its simplicity and reduction in the bias in the case of MOKA agents.
4) I like the presentation quality in the various plots and tables. In general, the writing quality is also maintained high throughout.

**Weaknesses:**

**Major (my reasons for not providing higher score):**

1. Lack of motivation behind use of particular metrics for bias calculation. Why do you use coefficient of variation? This metric would unnecessarily aggravate the impact of success rate variance in harder tasks. Personally, it would be more desirable to see how performance changes in the tasks where the success rate was already high. This would show that the tasks that were easy enough for the model become very hard if color or camera pose changes. So, an actionable change would be to introduce other metrics that disentangle average performance from the variance in the success. One such metric would be standard deviation.
2. Similarly, the interaction effect coefficient can be complemented with measurements of the covariance in the success rate as two entities change.
3. Figure 3:  What to interpret here? This plot can certainly be improved. Maybe classify these poses into certain broad name categories. Then identify the pattern where a pose would lead to failure. Contrast the patterns where different models
4.  Figure 4:  The font size of the x and y labels here is too small. The caption does not mention how to interpret the figure.
5. Issue in bias reduction in other models: The proposed bias reduction method does not lead to reduction in 2/3 models. For these agents, there is almost no change in the bias.

**Minor:**
1. It would be a good idea to describe any special model referred in the paper (e.g. MOKA) before using it to assert claims.
2. Section 3 is essentially about how to create combinations for analysis. It could have been explained in simpler language.

**Questions:**

1.  Of the different camera poses considered, how many are standard poses in VLM-based robotics policies? Among those typical poses, which ones exhibit significant bias? Are the VLM agents in general able to perform well under these typical poses?
2. You mention that “A key finding is that all agents have specific viewpoints that lead to complete task failure.” Can the authors comment on what kind of viewpoints are leading to such task failures?
3. Out of curiosity, how realistic is it that camera pose would change on a robot? Aren't cameras some of the best guarded equipments whose pose will not change drastically?

---

### Official Review · Reviewer_57MG · 2025-10-31

**Soundness:** 3
**Presentation:** 3
**Contribution:** 2
**Rating:** 4
**Confidence:** 3

**Summary:**

This paper presents RoboView-Bias, a benchmark for assessing visual bias in robotic manipulation agents. The benchmark is developed using a structured variant-generation framework, including visual perturbation factors (color, camera euler, camera poses, scale) and task context perturbation factors (initial position, geometric shapes and task instructions). The paper evaluates three agents, SimpleAgent (based on BadRobot), MOKA with two alternative LLMs (GPT4-o and Gwen-VL-Max) and pi zero. Results show that all agents are affected by color, while there are viewpoints that lead to task failure. There are also interaction effects between camera and pose. The authors also explore the idea of a Semantic Grounding Layer, which improves the bias of MOKA, but not other models.

**Strengths:**

- The paper investigates a timely issue of high relevance and importance, and does so using a systematic approach that investigates both the effect of individual parameters (visual and contextual factors), as well as their interaction.

 - The "perceptual fairness" validation step is interesting and supports the overall rigorous approach followed in the design of the benchmark.

 - The paper offers novel findings regarding the biases of the evaluated models, including color bias as well as interactions between color and camera pose.

**Weaknesses:**

- The benchmark concerns a single task and synthetic / simulated data. This raises the question on whether the results and observation also transfer to real-world data and/or other tasks.

- Additional factors would be expected for visual bias assessment, such as texture / patterns / objects and, more generally, complex visual elements that are closer to the real world.

 - The evaluation performed using the benchmark could be more extensive. The number of backbones and model types evaluated is limited. Furthermore, the analysis reports averages of metrics, without reporting metrics of statistical significance. For example for the $\pi_0$ model, could the observed differences for different colors be due to chance? Which of the pairwise differences are significant?

 - The SGL approach is based on heuristics, and does not offer a general-purpose  method for addressing this problem. Moreover, it does not lead to improvements in the Simple and $\pi_0$ models

  - More generally, it seems that the benchmark is in the right direction of addressing an important issue, but it seems too limited in scope to be useful in practical applications

**Questions:**

- Why doesn't SGL help with Simple and $\pi_0$?
- Do you have ways to disentangle the source of bias (architecture, underlying VLMs)?

---

### Official Review · Reviewer_ZJTK · 2025-11-01

**Soundness:** 2
**Presentation:** 3
**Contribution:** 2
**Rating:** 2
**Confidence:** 3

**Summary:**

This paper introduces RoboView-Bias - a benchmark designed to evaluate visual biases in robotic policies by factorizing various forms of visual biases and doing controlled experiments. The paper identifies camera viewpoint and bias towards high-saturation colors as biases in the methods evaluated.

**Strengths:**

1. This paper finds some important ways in which VLAs are biased. For example, the effect of low-saturation colors and the fact that most methods have some sort of a preferred camera pose are helpful observations.
2. The two-stage validation process in Perceptual Validation Pipeline is instructive in how evaluation pipelines should be tested and confounders should be handled before being deployed.
3. The case study of VLM-biases in high-level planning for MOKA is interesting - especially, the emphasis on how success rate is affected by the inconsistency of the labels used for describing the same object, even if they are all somewhat accurate.

**Weaknesses:**

1. The most important weakness of this paper, for me, is that it claims that the study of visual biases on robotic policies has not been done before and that RoboView-Bias is the first benchmark that aims to do so. However, there are quite a few papers that have done it. Gao et. al. introduced the Star-Gen framework that actually evaluates on almost all the metrics discussed in this paper. In particular, the factorized analysis is also presented in Star-Gen. Similar Wang et. al.'s VLATest is another paper to take a look at.
2. The types of visual bias considered here are quite narrow. There are several other important factors, such as lighting and distractor objects, that should be studied as well.
3. The choice of agents that are evaluated is not well-justified. There are four instantiations of VLM-driven agents (two for SIMPLE and two for MOKA) whereas only one (i.e. $\pi_0$) VLA policy being evaluated, even though VLAs, like openVLA, are known to be more reliable than methods like MOKA. It would be helpful to see another VLA being evaluated in order to also validate the observations from line 319 to 323.
4. The study of SGL needs to be significantly more detailed and ablated. It is not clear to me why the approach does not perform too well with $\pi_0$ and SimpleAgent. Without this being addressed, it is unclear what the contribution of SGL is to the paper from just observing its effect on MOKA. Similarly, Figure 7 is unclear in terms of what visual perturbation dimension the bias coefficient is corresponding to. Modularizing that would not just be helpful for clarity but also in helping us understand what SGL contributes. Finally, doing an ablation over multiple VLMs (say, between Qwen-VL-Max and GPT-4o) is necessary to understand the general strengths of SGL, because, as of now, it is unclear whether this result in Figure 7 is overfitted to a specific choice of a VLM.

Minor:
1. Please follow the citation protocol described in the latex template (using \citep{...} versus \citet{...}) - there are issues with spacing between citations and the rest of a line (for e.g. the first sentence of the introduction).

[1] Jensen Gao, Suneel Belkhale, Sudeep Dasari, Ashwin Balakrishna, Dhruv Shah, Dorsa Sadigh. A Taxonomy for Evaluating Generalist Robot Policies.
[2] Zhijie Wang, Zhehua Zhou, Jiayang Song, Yuheng Huang, Zhan Shu, Lei Ma. VLATest: Testing and Evaluating Vision-Language-Action Models for Robotic Manipulation.

**Questions:**

I am curious about how the two-stage validation pipeline worked in practice. How many iterations of the refinement were needed in stage 1 and stage 2?

---

### Official Review · Reviewer_ARmd · 2025-11-03

**Soundness:** 2
**Presentation:** 3
**Contribution:** 2
**Rating:** 2
**Confidence:** 4

**Summary:**

This paper introduces RoboView-Bias, a benchmark systematically quantify visual bias in embodied agents for robotic manipulation. The authors propose a structured variant-generation framework (SVGF) that isolates visual perturbation factors (color, camera viewpoint, scale) from task context factors to enable controlled bias measurement. They evaluate three representative agents across two paradigms (VLM-driven and VLA models) and report significant visual biases, with camera viewpoint being the most critical factor. The paper also proposes a Semantic Grounding Layer (SGL) as a mitigation strategy. The evaluation reveals strong asymmetric coupling between color and viewpoint biases, with viewpoint changes amplifying color-related bias more than vice versa.

**Strengths:**

- The paper provides interesting insights in understanding visual bias in embodied manipulation agents.
- The factor isolation principle through SVGF is well-motivated and enables attributable bias measurement.

**Weaknesses:**

- The benchmark only evaluates a single, simple grasping task where the robot picks up one object. It does not consider other skills such as pushing, sliding, insertion, assembly, and does not deal with more complex objects such as multi object and articulated object manipulation, deformable object handling. In the current simple task, precise object geometry, contact points, and spatial relationships are less critical. More complex tasks requiring fine-grained visual understanding (e.g., peg-in-hole, cable routing, multi-object assembly) may reveal entirely different bias patterns.
- All experiments are simulation-only, there is no real world evaluation, however, visual bias in simulation may not correlate with real-world failures.
- The semantic grounding layer works well only for MOKA but has minimal improvements for SimpleAgent and $\pi_0$. This further reduces the technical contribution of this work.

**Questions:**

- Why does SGL fail for SimpleAgent and π₀? The explanation that tasks are "too simplistic" seems contradictory. Can you provide quantitative analysis of when/why SGL works?
- Is there evidence that reducing bias (as measured by CV) actually improves overall task performance and safety? Could you show that low-bias agents generalize better to new scenarios?

---

### Meta-Review · Area_Chair_YJcF · 2026-01-02

**Summary:**

I recommend reject for this paper, informed by the following reviewer feedback (many of the concerns are shared across reviewers) and the lack of any rebuttal or author discussion:

**ARmd** states that the *strengths* are the proposed structured variant-generation framework (SVGF) and the insights that come from it. The main *concerns* are that the evaluations are only in simulation with a single task (e.g., picking up one object) and the proposed semantic grounding layer provides marginal gains over baselines.

**ZJTK** states that the *strengths* are the case studies on VLA bias and what sorts of biases exist. The main *concerns* are the lack of novelty (there are several existing works studying the visual biases in VLAs, cited by the reviewer), the narrowness of the visual bias under study, under-justified use of agents, and the proposed semantic grounding layer provides marginal gains over baselines.

**57MG** states that the *strengths* are the timely nature of the problem and some relevant findings about bias. The main *concerns* are that the evaluations are only in simulation with a single task (e.g., picking up one object), the narrowness of the visual bias under study and evaluation, and the heuristic nature of the semantic grounding layer.

**dtBd** states that the *strengths* are the paper writing, the relevant findings about bias, and the simplicity of the semantic grounding layer. The main *concerns* are about the lack of justification for the metrics, the lack of significance of the results over baselines, and clarity in the figures.

**RXBV** states that the *strengths* are the strong motivation,  the proposed structured variant-generation framework (SVGF) and the insights that come from it. The main *concerns* are the apparent heavy use of LLM writing, the mathematical notation that should be written more carefully in order to actually clarify the formalism rather than obfuscate it, limited evaluation, and the proposed semantic grounding layer provides marginal gains over baselines.

**Reviewer Concerns:**

There was no rebuttal or author discussion with the reviewers, so all the concerns listed above are still outstanding.

**Reviewer Scores:**

* **ARmd** would have *maintained* a score of 2: reject, not good enough
* **ZJTK** would have *maintained* a score of 2: reject, not good enough
* **57MG** would have *maintained* a score of 4: marginally below the acceptance threshold.
* **dtBd** would have *maintained* a score of 6: marginally above the acceptance threshold.
* **RXBV** would have *maintained* a score of 2: reject, not good enough

---

### Decision · Program_Chairs · 2026-01-26

Reject